# Low-Carbohydrate Diets and Mortality in Older Asian People: A 15-Year Follow-Up from a Prospective Cohort Study

**DOI:** 10.3390/nu14071406

**Published:** 2022-03-28

**Authors:** Ce Sun, Wei-Sen Zhang, Chao-Qiang Jiang, Ya-Li Jin, Xue-Qing Deng, Jean Woo, Kar-Keung Cheng, Tai-Hing Lam, G. Neil Thomas, Lin Xu

**Affiliations:** 1School of Public Health, Sun Yat-sen University, Guangzhou 510080, China; sunc5@mail2.sysu.edu.cn (C.S.); dengxq@mai.sysu.edu.cn (X.-Q.D.); 2Guangzhou Twelfth People’s Hospital, Guangzhou 510620, China; zwsgzcn@163.com (W.-S.Z.); jcqianggz@163.com (C.-Q.J.); jinyali22@163.com (Y.-L.J.); hrmrlth@hku.hk (T.-H.L.); 3Department of Medicine and Therapeutics, The Chinese University of Hong Kong, Hong Kong, China; jeanwoowong@cuhk.edu.hk; 4Institute of Applied Health Research, University of Birmingham, Birmingham B15 2TT, UK; k.k.cheng@bham.ac.uk; 5School of Public Health, The University of Hong Kong, Hong Kong, China

**Keywords:** low-carbohydrate diet, mortality, older people, diabetes

## Abstract

The long-term effects of a low-carbohydrate diet (LCD) on mortality, accounting for the quality and source of the carbohydrate, are unclear. Hence, we examined the associations of LCDs with all-cause and cause-specific mortality in a prospective cohort study. A total of 20,206 participants (13.8% diabetes) aged 50+ years were included. Overall, vegetable-based and meat-based LCD scores were calculated based on the percentage of energy as total and subtypes of carbohydrates, fat, and protein. Cox regression analysis was used to calculate hazard ratios (HRs) and 95% confidence intervals (CIs). During 294,848 person-years of follow-up, 4624 deaths occurred, including 3661 and 963 deaths in participants without and with diabetes, respectively. In all participants, overall LCD score was not associated with all-cause and cause-specific mortality, after multivariable adjustment. However, for the highest versus the lowest quartiles of vegetable-based LCD, the adjusted HRs (95%CIs) of all-cause and CVD mortality were 1.16 (1.05–1.27) and 1.39 (1.19–1.62), respectively. The corresponding values for highest versus lowest quartiles of meat-based LCD for all-cause and CVD mortality were 0.89 (0.81–0.97) and 0.81 (0.70–0.93), respectively. Similar associations were found in participants without diabetes. In patients with diabetes, the adjusted HR (95%CI) of CVD mortality for the highest versus the lowest quartiles of vegetable-based LCD was 1.54 (1.11–2.14). Although there were no significant associations with overall LCD score, we found that the vegetable-based LCD score was positively, whereas the meat-based LCD score was negatively, associated with all-cause and CVD mortality in older Asian people.

## 1. Introduction

The primary source of energy from food worldwide is carbohydrates, providing over 50% of the daily energy intake, followed by sources from fat and protein [1]. However, there are substantial differences in the proportion of macronutrient intakes between Asia and Western countries [2,3]. The traditional Chinese diet is characterized by a high intake of carbohydrates and vegetables, as well as moderate intake of animal foods [4]. In a recent meta-analysis of predominantly Western populations (six of seven studies), high carbohydrate intake was associated with a higher risk of all-cause mortality and cardiovascular disease (CVD) [5]. Although low-carbohydrate diets (LCD) have been suggested to be useful in weight, blood pressure, and glucose management in short-term randomized controlled trials [6,7], studies on the long-term health effects of LCD remain controversial [8], probably due to different levels and sources of baseline carbohydrate intake in diverse populations. In addition, we found only one study in Asia showing a U-shaped association between LCD score and total mortality in adults (age range: 40–69 years) [9]. Since this study was launched in 1990 and 1993, and there has been a substantial worldwide improvement in nutrition over the past two decades, more contemporary evidence is needed.

In addition to the quantity of carbohydrate, the quality and source might also play a role in health outcomes [1,10]. Carbohydrates from refined grains and added sugar were associated with a higher risk of diabetes and CVD, whereas those from whole grains, non-starchy vegetables, and fruits were associated with a lower risk [1,11]. Likewise, high-quality carbohydrates (whole grains, non-starchy vegetables and fruits) or low-carbohydrate diets are recommended for patients with diabetes to manage glycemic index and glycemic load, although long-term compliance is low [12,13]. Despite the increasing popularity of LCD diets, we searched PubMed using the keywords “low carbohydrate diet” or “carbohydrate quality” and “mortality” up to 15 March 2022, and found no population-based cohort studies reporting the association of LCDs with mortality by considering both quality and sources of carbohydrates in Asians and non-Asians. In addition, a previous study showed that substituting unsaturated fat for saturated fat [14] or substituting plant protein for animal protein [15] was associated with a lower risk of CVD mortality in Western countries, indicating that the sources of protein and fat might also play a role in health outcomes.

We therefore conducted a prospective cohort study, using data from Guangzhou Biobank Cohort Study (GBCS), to investigate the associations of types of LCD (total, meat-based, and plant-based LCD) with the risk of all-cause, cancer, and CVD mortality in an older sample. Furthermore, we also examined whether the associations of types of LCD with the risk of all-cause, cancer, and CVD mortality varied by diabetes status.

## 2. Methods

### 2.1. Study Design and Sample

The GBCS is a population-based cohort study in South China [16]. Briefly, GBCS is a three-way collaboration among the Guangzhou Twelfth People’s Hospital and the Universities of Hong Kong and Birmingham. All participants were recruited from a community social and welfare association, the Guangzhou Health and Happiness Association for the Respectable Elders (GHHARE), from 2003 to 2008. GHHARE is a large, unofficial organization with ten branches throughout all districts of Guangzhou. Membership in this association is open to Guangzhou residents aged 50 years or older for a nominal, monthly fee of CNY 4 (≈USD .50). Baseline information was collected using computer-assisted face-to-face interviews by trained nurses. Information on anthropometrics, blood pressure, fasting plasma glucose, lipids, and inflammatory markers was collected following standard protocols. The reliability of the questionnaire was tested 6 months into recruitment by recalling 200 randomly selected participants for re-interview, and the results were satisfactory [16]. Ethics approval was granted by the Guangzhou Medical Ethics Committee of the Chinese Medical Association, Guangzhou, China (IRB No. GWYL-2019-125). As the Food Frequency Questionnaire (FFQ) was shortened in phase 3 of the baseline (2006–2008), participants from phase 3 were not included in the current analysis.

### 2.2. Assessment of LCD Score

Information on diet was collected using a FFQ validated by Woo et al. [17]. The LCD diet score was calculated as per the method described in a recent study by Shan et al. [8]. Briefly, percentages of energy from carbohydrate, fat, and protein for each participant were each calculated and used to rank the participants into 11 strata. For carbohydrates, participants in the lowest group received 10 points and those in the highest group received 0 points. The order of the strata for fat and protein was reversed. The scores of the three macronutrients were summed to create an overall LCD score, which ranged from 0 to 30. Two additional LCD scores were also created: (1) vegetable-based LCD scores were calculated according to the percentage of energy from high-quality carbohydrates, plant protein, and unsaturated fat; (2) meat-based LCD scores were calculated according to the percentage of energy from low-quality carbohydrates, animal protein, and saturated fat (Appendix A). Based on the Healthy Eating Index (HEI) 2015, high-quality carbohydrate was defined as a carbohydrate from whole grains, whole fruits, legumes, and non-starchy vegetables, and low-quality carbohydrate as a carbohydrate from refined grains, added sugar, fruit juice, potatoes, and other starchy vegetables [3]. As we found a significant interaction between LCDs and diabetes in all-cause mortality (*p* for interaction < 0.001), we also conducted pre-specified analyses by diabetes status (Appendix A). Diabetes was defined by having a history of diabetes or fasting glucose ≥7.0 mmol/L at baseline.

### 2.3. Ascertainment of Mortality

Details of the methods were described elsewhere, and information on the causes of death up to 19 April 2021 was obtained through record linkage with the Death Registry of the Guangzhou Center for Disease Control and Prevention (GCDC) [18]. Briefly, causes of death were coded by trained nosologists in each hospital according to the International Classification of Diseases, Tenth Revision (ICD-10). If death certificates were not issued by medical institutions, the causes of death were verified by GCDC as part of its quality assurance program by cross-checking past medical history and conducting a verbal autopsy. Moreover, we also conducted verbal autopsy meetings in the Guangzhou Twelfth People’s Hospital to further clarify the deaths of unclear causes. In the present study, the primary outcome was mortality from all causes, and the secondary outcome was mortality from cancer and CVD.

### 2.4. Potential Confounders and Mediators

As sex, age, socioeconomic factors (education, family income) [19], lifestyle factors (smoking, drinking, and physical activity), body mass index (BMI) [20], and history of cancer and CVD were associated with both dietary pattern and mortality, these factors were considered as potential confounders and adjusted in the regression model. The potential mediators between LCD score and all-cause mortality risk included systolic blood pressure (SBP), fasting plasma-glucose (FPG), total cholesterol (TC), and self-rated health at baseline. Procedures for measuring these were reported previously [16].

### 2.5. Statistical Analysis

The chi-square test and one-way analysis of variance (ANOVA) were used to compare baseline categorical and continuous variables by quartiles of LCD scores, respectively. Person-years of follow-up were assessed from the date of baseline enrollment until death or the end of the present study on 19 April 2021, whichever came first. The LCD scores were categorized into quartiles. Multivariable Cox proportional hazards regression was used to calculate hazard ratios (HRs) and 95% confidence intervals (CI) of mortality associated with the LCD score. Schoenfeld residuals were used to test the proportional hazard assumption and no violations of the proportional hazard assumption were found. Model 1 was the crude model without any adjustment. In multivariable analyses, model 2 adjusted for sex and age, and model 3 additionally adjusted for education, family income, smoking, drinking, physical activity, BMI, and history of cancer and CVD. Model 4 adjusted for determinants considered potential mediators, namely, SBP, FPG, TC, and self-rated health. In addition, the non-linearity of the effect of the LCD score on mortality risk was estimated by adding a quadratic term to the model with the quartiles of LCD scores as a continuous variable, and the fitness of the models with and without the quadratic term was compared using the likelihood ratio (LR) test [21]. A non-significant *p*-value was interpreted as an indication of a linear effect of the LCD score on mortality risk.

Furthermore, a stratification analysis was performed for the associations between diet score and all-cause mortality according to several potential effect modifiers at the baseline. As many statistical tests were performed in the subgroup analysis, we used the Bonferroni correction to account for multiple testings, and the significance level was set at *p* < 0.002 (0.05/8 [subgroups] × 3 [dietary scores]). To assess the extent to which baseline risk factors explained the associations of the LCD score with mortality, the percentage of excess risk mediated (PERM) was calculated as PERM = [HR (E + C) − HR (E + C + M)]/[HR (E + C) − 1] × 100, where E = exposure (types of LCD score), C = covariates (sex, age, education, family income, smoking, drinking, BMI, physical activity, and history of cancer and CVD), and M = explanatory variable being tested [21]. The following four groups of explanatory variables, (1) SBP; (2) FPG; (3) TC; (4) self-rated health, were included the PERM model. Finally, all the explanatory variables were included in the same model simultaneously. To rule out potential bias due to reverse causality (i.e., disease pathology— possibly subclinical—having an adverse impact on both dietary pattern and survival), we conducted a sensitivity analysis excluding participants who died within the first three years of follow-up. Statistical analysis was performed using Stata (Statacorp LP, version 15). Two-sided *p*-values < 0.05 were considered as statistically significant.

## 3. Results

### 3.1. Participant Characteristics

Of 20,490 participants, 128 with potentially unreliable dietary intake (<800 or >4200 kcal/d in men, and <600 or >3500 kcal/d in women), 57 who were followed up for less than 1 year, and 99 lost to follow-up with unknown vital status were excluded (Appendix A). A total of 20,206 participants, mean (SD) age = 62.7 (6.7) years; 14,423 (71.4%) women, including 17,416 participants without diabetes, mean (SD) age = 62.5 (6.7) years; 12,364 (71.0%) women, and 2790 participants with diabetes, mean (SD) age = 64.1 (6.2) years; 2059 (73.8%) women, were included in the present analysis. During an average of 14.8 years (SD = 3.3) with 294,848 person-years of follow-up, 4624 deaths occurred, including 1534 from cancer, 1783 from CVD, and 1307 from other causes. 

Table 1 shows that compared with a low LCD score (Q1), a high LCD score (Q4) was associated with being a woman and having a younger age, a higher education level and family income, lower physical activity, never smoked, and being a current alcohol consumer (all *p* < 0.05). Moreover, the potential mediators, lower FPG, SBP level, and good/very good self-rated health were found in those with a higher overall LCD score (*p* < 0.05). Similar results were found in those with a higher meat-based LCD score, although the BMI was lower, but with an increased history of cancer and CVD. In contrast, those with a higher vegetable-based LCD score had a lower education level, with a greater proportion of smokers and higher SBP and lower TC levels. A higher vegetable-based LCD score showed no association with sex, age, family income, drinking, BMI, or history of CVD. Participants without diabetes showed similar patterns to all participants (Appendix A). In contrast, participants with diabetes showed no association of overall LCD score with sex, age, smoking, drinking, and FPG level, no association of vegetable-based LCD score with smoking, history of cancer, and SBP level, and no association of meat-based LCD score with age, BMI, history of CVD, and FPG level, but a negative association between meat-based LCD score and drinking (Appendix A).

### 3.2. Mortality and LCD Score

Table 2 shows that in all participants, after adjusting for sex, age, education, family income, smoking, drinking, physical activity, BMI, and history of cancer and CVD, the overall LCD score showed no association with all-cause mortality. For the vegetable-based LCD score, the adjusted HR (95% CI) of all-cause mortality for the 2nd, 3rd, and 4th quartile versus the 1st quartile (Q1), was 0.99 (0.91–1.07), 1.11 (1.02–1.21), and 1.16 (1.05–1.27) (*p* for trend <0.001 and *p* for non-linear = 0.18), respectively. For the meat-based LCD score, the adjusted HR (95% CI) of all-cause mortality for Q2, Q3, and Q4 versus Q1 was 0.89 (0.83–0.97), 0.90 (0.83–0.97), and 0.89 (0.81–0.97) (*p* for trend = 0.007 and *p* for non-linear = 0.06), respectively. 

Appendix A shows no association between overall LCD score and all-cause mortality in participants with or without diabetes. In those without diabetes, the results of the vegetable-based LCD and meat-based LCD scores were generally similar to those from the total population. Comparing with the Q1 group, participants in Q4 of the vegetable-based LCD scores showed a higher risk of all-cause mortality (HR 1.10, 95% CI 1.01–1.23), whereas those with the highest quartile of meat-based LCD scores showed a lower risk of all-cause mortality (HR 0.87, 95% CI 0.79–0.97). In participants with diabetes, no associations of the vegetable-based LCD score and meat-based LCD score quartiles with all-cause mortality were found, although there was a linear trend between the vegetable-based LCD score and all-cause mortality (*p* for trend = 0.04).

Appendix A shows no association of the overall LCD score with mortality from cancer, CVD, and other causes. The vegetable-based LCD score was associated with a higher risk of CVD mortality (Q1: reference, Q2: 1.18 (1.03–1.34), Q3: 1.36 (1.18–1.56), and Q4: 1.39 (1.19–1.62), *p* for trend < 0.001 and *p* for non-linear = 0.15), whereas the higher meat-based LCD score quartiles were associated with a lower risk of CVD mortality (Q1: reference, Q2: 0.84 (0.75–0.95), Q3: 0.82 (0.72–0.93), and Q4: 0.81 (0.70–0.93), *p* for trend = 0.02 and *p* for non-linear = 0.10). Appendix A shows similar results in participants without diabetes. In participants with diabetes, no association of overall LCD score and meat-based LCD score with CVD mortality was found. However, we found a positive association between vegetable-based LCD score quartiles and CVD mortality (Q1: reference, Q2: 1.21 (0.89–1.63), Q3: 1.59 (1.18–2.15), Q4: 1.54 (1.11–2.14), *p* for trend = 0.003 and *p* for non-linear = 0.25).

Figure 1 shows that the HR of all-cause mortality comparing Q4 to Q1 of vegetable-based LCD scores was 1.16 (1.05–1.27) after adjustment for potential confounders. The HR decreased by 14% after adjusting for SBP, 27% after adjusting for FPG, and 2% after adjusting for self-rated health, and increased by 3% after adjusting for TC. The overall attenuation after adjustment for mediators was 41%. Similar patterns were found for the association between vegetable-based LCD scores and cause-specific mortality. FPG appeared to be the strongest mediator in a vegetable-based LCD diet. For a meat-based LCD, the HR of all-cause mortality was 0.89 (0.81–0.97) after adjustment for potential confounders, which increased by 10% after adjusting for SBP, 24% after adjusting for FPG, and 1% after adjusting for self-rated health. FPG appeared to be the strongest mediator in a meat-based LCD diet. 

Figure 2 shows that in participants without diabetes, the vegetable-based LCD score was associated with a higher risk of mortality from all-cause mortality (HR comparing Q4 to Q1 = 1.10, 95% CI 1.01–1.23) and CVD (HR 1.26, 95% CI 1.06–1.51). After adjustment for mediators, the HRs of all-cause mortality became non-significant (HR 1.06, 95% CI 0.96–1.18). SBP appeared to be the strongest mediator (PERM = 26% for all-cause mortality and 19% for CVD mortality). In participants with diabetes, no association of the vegetable-based LCD score with all-cause mortality was found. However, we found that the vegetable-based LCD score was associated with a higher risk of CVD mortality (adjusted HR 1.54, 95% CI, 1.11–2.13), for which TC appeared to be the strongest mediator (PERM = 16%). Figure 3 shows that in participants without diabetes, the meat-based LCD score was associated with a lower risk of all-cause and CVD mortality, which was partly mediated by SBP (PERM = 13% for all-cause and CVD mortality). No association of the meat-based LCD score with cancer and other-cause mortality was found.

### 3.3. Subgroup and Sensitivity Analyses

Appendix A show similar associations in most subgroups. After Bonferroni corrections for multiple testing, the association between the vegetable-based LCD score and all-cause mortality appeared to be stronger in obese than non-obese participants (Appendix A, *p* for interaction <0.001). A higher vegetable-based LCD score was associated with a higher risk of all-cause mortality (HR for Q4 versus Q1 = 1.55, 95% CI 1.18–2.04). Similar results were observed in participants without diabetes (Appendix A, all *p* for interaction <0.001). Similar results were also found after excluding deaths within the first three years of follow-up (Appendix A). 

## 4. Discussion

After a long-term follow-up for nearly 15 years, no association of overall LCD scores with the risk of all-cause and cause-specific mortality was found in our study. In prespecified subgroup analyses, however, we found that the vegetable-based LCD score was positively, whereas the meat-based LCD score was negatively, associated with all-cause and CVD mortality in older Asian people. Similar associations were observed for participants without diabetes. In participants with diabetes, a positive association of the vegetable-based LCD score with the risk of CVD mortality was found. 

### 4.1. Comparison with Previous Studies

Most studies considered LCD based on animal-derived protein and fat sources as a risk factor of mortality, whereas an LCD based on plant-derived protein and fat reduced mortality [5,22,23]. Furthermore, studies show that higher levels of whole grain intake were associated with a lower risk of all-cause mortality, whereas refined grain intake was associated with a higher risk of all-cause mortality [24,25]. This highlights that a healthy LCD diet is not only dependent on the sources of macronutrients, but also on the quality of these nutrients. A previous study using a new classification approach for LCD scores found that participants with low low-quality carbohydrate, high unsaturated fat, and high plant protein intake had a lower all-cause and cancer mortality risk, while those with a low high-quality carbohydrate and high saturated fat and animal protein intake had a higher all-cause mortality risk [8]. Our results generally supported the intake of high-quality carbohydrate, and further showed that participants with low low-quality carbohydrate and high saturated fat and animal protein intake had lower all-cause and CVD mortality risk, and those with low high-quality carbohydrate and high unsaturated fat and plant protein intake had a higher mortality risk. This discrepancy might be due to the differential amount and sources of carbohydrate, fat, and protein in the Western and Eastern diets. 

The percentage of energy from carbohydrates, fat, and protein in our study were similar with the results of China Health and Nutrition Survey (CHNS) [26]. Notably, the percentage of energy from carbohydrate (especially high-quality carbohydrate) in our study was higher than that reported in the US (total carbohydrate, 57.1% versus 50.5%; high-quality carbohydrate, 10.6% versus 8.6%, respectively), whereas the percentage of energy from animal protein and saturated fat intake was much lower than the US (animal protein, 7.4% versus 10.4%; saturated fat, 4.9% versus 11.9%, respectively) [3]. Moreover, compared with the US, total per capita consumption of meat in Asians was much lower (49.4 kg/year versus 122.8 kg/year), whereas the percentage of energy from fish/sea food consumption was higher (43.5% versus 26.0%) [27]. Some recent studies showed that fish/seafood consumption was associated with a lower risk of all-cause and CVD mortality in Asians, but not in the US populations [28,29]. Meta-analyses show that total mortality is higher in participants who have high intakes of both red and processed meat than in those with low meat intakes in Western high-income countries [30]. However, meat is good source of energy, as well as a range of essential nutrients, including protein and micronutrients such as iron, zinc, and vitamin B12 for low-income countries. A previous study showed that Indian vegetarians had a more favorable cardiovascular risk profile than did non-vegetarians [31]. Along with these findings, our results support the beneficial effects of moderate consumption of animal protein. In addition, the non-significant association between meat-based LCD and CVD mortality in patients with diabetes could also be explained by the lower levels of fish consumption in the diabetes group compared to those without diabetes [28]. A recent meta-analysis also showed that substituting fish with red and processed meat was associated with increased risks of all-cause mortality in patients with type 2 diabetes [32]. 

Apart from differential amounts of high-quality carbohydrates, saturated fat, and animal protein intake, the discrepancies could also be at least partly explained by the low-quality carbohydrate, unsaturated fat, and plant protein consumed. Notably, compared with the US, the percentage of energy from low-quality carbohydrate (46.4% versus 41.8%) and plant protein (8.5% versus 5.8%) intake was higher in our sample, whereas the percentage of energy from unsaturated fat was lower (monounsaturated fatty acids, 8.6% versus 13.1%; polyunsaturated fatty acids, 6.3% versus 8.2%) [3]. Regarding the results on the vegetable-based LCD, our results were generally consistent with previous studies in Asia showing positive associations between plant-based diets consisting of a high intake of refined carbohydrates and the risk of metabolic syndrome and CVD [33,34]. In our study, participants with a higher vegetable LCD score had higher levels of unsaturated fat consumption and higher risks of all-cause and CVD mortality, which could be partly explained by the cooking method. In traditional Chinese cuisine, plant oil is often used for stir-frying, pan-frying, and deep-frying, and it is heated to a high temperature [35]. High heat has been shown to cause partial hydrogenation of unsaturated plant oils to produce trans fats. Studies have consistently shown trans fats consumption to be associated with a higher risk of all-cause and CHD mortality [36]. Moreover, as CVD is a leading cause of mortality in people with type 2 diabetes mellitus [37], the stronger positive association between vegetable-based LCD and CVD mortality in participants with diabetes in our study also warrants further attention.

Regarding the null association between the overall LCD and mortality, our results were consistent with some [8,38] but not all [9,22] previous studies. For example, a recent study in Japan showed a U-shape association between overall LCD score and all-cause mortality [9]. The authors suggested that the sources of food might have modified the association [9]. Another study in the US showed a positive association between overall LCD and all-cause mortality [22]. The differences in the results could be partly due to the substantial variation in carbohydrate consumption across different populations (i.e., about 60% of the overall energy was from carbohydrate in Asians vs. 50.5% in the US) [2,3] and accounting for the quantity of carbohydrate intake. Higher low-quantity carbohydrate consumption could lead to a greater glycemic burden, and a subsequently higher risk of insulin resistance and vascular complications [39,40,41], which warrants further research in populations with a high carbohydrate intake. 

In addition, we found that the meat-based LCD score was associated with a lower risks of all-cause and CVD mortality, which was generally consistent with the results of a previous study showing a beneficial effect of higher animal protein intake on all-cause mortality in older Chinese men [42]. However, such association was not evident, and even inverse, in a previous general population study in the US [43]. As previous population-based cohort studies did not report the association of LCDs with mortality, accounting for both quality and sources of carbohydrates in older people, our study adds to the literature by showing the long-term health effects related to LCD patterns and its subtypes in older Asians.

### 4.2. Strengths and Limitations 

The strengths of our study included the large sample size, long duration of follow-up, and the comprehensive adjustment for potential confounders. However, there were some limitations in the present study. First, changes in dietary patterns were not assessed during follow-up. However, our previous study found that the dietary patterns of our sample were relatively stable [44,45]. Second, residual confounding could not be fully ruled out, although we adjusted for a wide range of potential confounding factors reported in previous literature. Third, our results may not be directly applicable to younger or Western populations. Fourth, the null association in the subgroup of participants with diabetes could be due to the relatively small sample size. Although a recent meta-analysis showed that patients adhering to an LCD for six months may experience remission of diabetes without adverse consequences [46], further studies on the health effects related to long-term and types of LCD patterns in patients with diabetes are warranted.

## 5. Conclusions

In this study of older Asian people, overall LCD score was not associated with all-cause or cause-specific mortality. However, we found that the vegetable-based LCD score was positively, whereas meat-based LCD score was negatively, associated with all-cause and CVD mortality in older Asian people. Inconsistencies in the literature on the health effects of an LCD may reflect the importance of the local diet and age-related nutrient composition of the diet.

## Figures and Tables

**Figure 1 nutrients-14-01406-f001:**
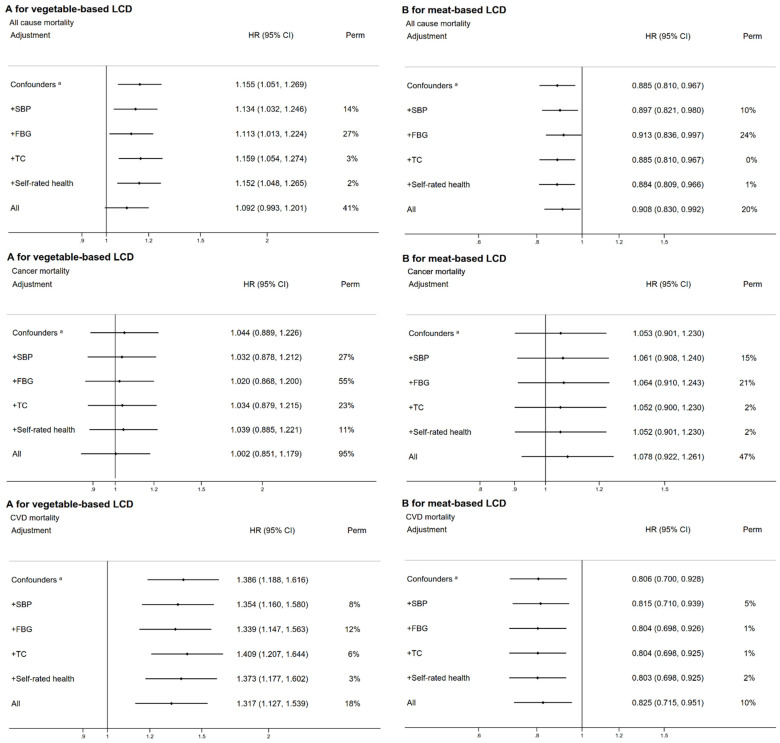
The associations between the LCD score (Q4 versus Q1) and all-cause mortality, and the proportions of the associations attributable to systolic blood pressure, fasting plasma-glucose, total cholesterol, and self-rated health in all-participants (**A**: vegetable-based LCD score, **B**: meat-based LCD score).

**Figure 2 nutrients-14-01406-f002:**
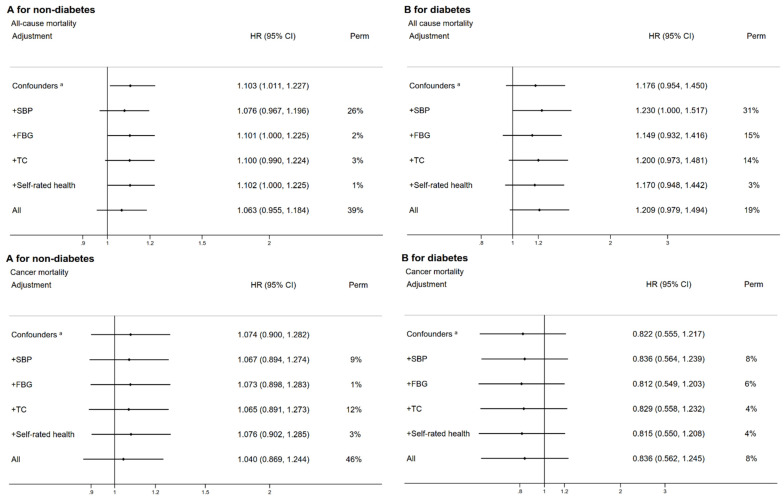
The associations between the vegetable-based LCD score (Q4 versus Q1) and mortality, and the proportions of the associations attributable to systolic blood pressure, fasting plasma-glucose, total cholesterol, and self-rated health (**A**: participants without diabetes, **B**: participants with diabetes).

**Figure 3 nutrients-14-01406-f003:**
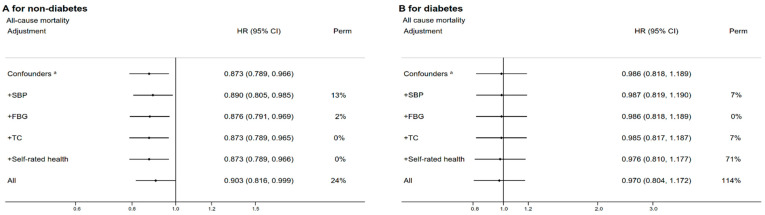
The associations between the meat-based LCD score (Q4 versus Q1) and mortality, and the proportions of the associations attributable to systolic blood pressure, fasting plasma-glucose, total cholesterol, and self-rated health (**A**: participants without diabetes, **B**: participants with diabetes).

**Table 1 nutrients-14-01406-t001:** The baseline characteristics by quartiles of low-carbohydrate-diet scores in 20,206 participants recruited from 2003–2008 ^a^.

Characteristic	Overall Low-Carbohydrate-Diet Score		*p*-Value	Vegetable-Based Low-Carbohydrate-Diet Score		*p*-Value	Meat-Based Low-Carbohydrate-Diet Score		*p*-Value
	Quartile 1	Quartile 4		Quartile 1	Quartile 4		Quartile 1	Quartile 4	
Number of participants	5100	4936		5278	3775		5163	4418	
Age, mean (SD), year	63.5 (6.5)	62.1 (6.8)	0.04	62.7 (6.7)	62.8 (6.6)	0.70	63.4 (6.6)	62.0 (6.8)	0.02
Sex			<0.001			0.56			<0.001
Women	1696 (33.3)	3707 (75.1)		3740 (70.9)	2677 (70.9)		3398 (65.8)	3432 (77.7)	
Men	3404 (66.7)	1229 (24.9)		1538 (29.1)	1098 (29.1)		1765 (34.2)	986 (22.3)	
Education level			<0.001			<0.001			<0.001
Less than primary school	2678 (52.5)	1988 (40.3)		2231 (42.3)	1878 (49.8)		2826 (54.8)	1619 (36.6)	
Middle school	2009 (39.4)	2445 (49.5)		2463 (46.7)	1609 (42.6)		2003 (38.8)	2305 (52.2)	
College or above	413 (8.1)	503 (10.2)		584 (11.0)	288 (7.6)		330 (6.4)	493 (11.2)	
Family income, RMB/year			<0.001			0.21			<0.001
<20,000	1261 (24.7)	820 (16.6)		992 (18.8)	764 (20.2)		1297 (25.2)	688 (15.6)	
20,000–30,000	986 (19.3)	935 (18.9)		1062 (20.1)	756 (20.0)		1004 (19.5)	839 (19.0)	
30,000–50,000	781 (15.3)	1063 (21.5)		979 (18.6)	689 (18.3)		812 (15.7)	943 (21.4)	
≥50,000	473 (9.3)	902 (18.3)		768 (14.5)	550 (14.6)		443 (8.6)	880 (20.0)	
Do not know	1599 (31.4)	1216 (24.7)		1474 (28.0)	1016 (26.9)		1598 (31.0)	1060 (24.0)	
BMI, mean (SD), kg/m^2^	23.8 (3.3)	23.7 (3.3)	0.71	23.7 (3.2)	23.8 (3.3)	0.63	23.9 (3.3)	23.7 (3.3)	0.02
Physical activity			<0.001			<0.001			<0.001
Low	305 (6.0)	479 (9.7)		285 (5.4)	498 (13.2)		378 (7.4)	397 (9.0)	
Moderate	2113 (41.4)	2630 (53.3)		2416 (45.8)	1808 (47.9)		2155 (41.7)	2347 (53.1)	
High	2682 (52.6)	1827 (37.0)		2577 (48.8)	1469 (38.9)		2630 (50.9)	1674 (37.9)	
Smoking			<0.001			<0.001			<0.001
Never	3944 (77.4)	4045 (82.0)		4301 (81.5)	2890 (76.6)		3915 (75.9)	3731 (84.5)	
Past	581 (11.4)	393 (7.9)		532 (10.1)	380 (10.0)		600 (11.6)	327 (7.4)	
Current	575 (11.3)	498 (10.1)		445 (8.4)	506 (13.4)		648 (12.5)	360 (8.1)	
Drinking			0.001			0.24			0.004
Never	4139 (81.4)	3919 (79.5)		4170 (79.2)	3052 (81.0)		4164 (80.8)	3547 (80.5)	
Current	823 (16.1)	915 (18.5)		991 (18.7)	639 (16.9)		869 (16.8)	785 (17.7)	
Past	138 (2.6)	102 (2.0)		116 (2.0)	84 (2.1)		130 (2.4)	86 (1.8)	
Dietary intake, mean (SD)									
Total energy, kcal/d	1853 (520)	1730 (502)	0.003	1817 (530)	1783 (488)	<0.001	1835 (506)	1781.0 (520)	<0.001
Total carbohydrate, % of total energy intake	67.6 (4.7)	46.5 (5.4)	<0.001	60.9 (8.0)	52.2 (8.7)	<0.001	65.9 (6.1)	48.0 (7.4)	<0.001
High-quality carbohydrate	10.1 (7.3)	10.0 (6.1)	<0.001	15.1 (7.2)	5.6 (3.8)	<0.001	7.4 (4.4)	13.8 (8.5)	<0.001
Low-quality carbohydrate	57.4 (8.7)	36.4 (7.1)	<0.001	45.8 (10.2)	46.5 (9.7)	<0.001	58.4 (6.6)	34.0 (6.9)	<0.001
Total protein, % of total energy intake	14.2 (1.9)	18.0 (2.8)	<0.001	16.5 (2.9)	15.3 (3.1)	<0.001	14.0 (2.0)	18.3 (2.8)	<0.001
Animal protein	5.2 (1.8)	10.1 (2.9)	<0.001	8.1 (3.1)	6.8 (2.8)	<0.001	5.1 (1.7)	10.1 (2.9)	<0.001
Plant protein	9.0 (1.3)	7.9 (1.8)	<0.001	8.4 (1.6)	8.5 (1.8)	<0.001	8.9 (1.3)	8.2 (2.0)	<0.001
Total fat, % of total energy intake	12.9 (5.6)	26.1 (7.2)	<0.001	14.5 (6.0)	26.1 (7.8)	<0.001	13.2 (6.0)	26.6 (7.3)	<0.001
Saturated fat	3.4 (1.5)	6.2 (1.7)	<0.001	4.3 (1.8)	5.6 (1.8)	<0.001	3.0 (1.0)	7.0 (1.5)	<0.001
Monounsaturated fat	5.6 (2.5)	11.4 (3.4)	<0.001	6.2 (2.7)	11.4 (3.7)	<0.001	5.8 (2.7)	11.4 (3.7)	<0.001
Polyunsaturated fat	3.9 (2.4)	8.5 (3.5)	<0.001	4.0 (2.5)	9.2 (3.7)	<0.001	4.4 (2.8)	8.2 (3.7)	<0.001
History of CVD	2058 (40.8)	2.052 (42.1)	0.20	2171 (41.7)	1581 (42.4)	0.70	2013 (39.4)	1908 (43.7)	<0.001
History of cancer	98 (1.9)	108 (2.2)	0.63	136 (2.6)	57 (1.5)	0.005	98 (1.9)	104 (2.4)	0.004
Fasting plasma-glucose, mmol/L	5.9 (1.6)	5.7 (1.9)	<0.001	5.9 (1.6)	5.7 (2.0)	<0.001	5.8 (1.7)	5.8 (1.9)	<0.001
Systolic blood pressure, mmHg	133.4 (22.4)	130.1 (22.2)	<0.001	130.3 (22.2)	132.6 (22.3)	<0.001	133.4 (22.4)	129.6 (22.2)	<0.001
Total cholesterol, mmol/L	3.7 (1.0)	3.7 (0.9)	0.51	3.7 (1.0)	3.6 (1.0)	<0.001	3.7 (1.0)	3.7 (1.0)	0.90
Self-rated health									
Good/very good	4296 (85.1)	4409 (83.0)	0.02	4450 (85.5)	3096 (83.1)	0.004	4353 (85.2)	3604 (82.6)	0.005
Poor/very poor	752 (14.9)	827 (17.0)		758 (14.6)	632 (16.9)		754 (14.8)	760 (17.4)	

Abbreviations: BMI, body mass index; CVD, Cerebrovascular disease; SD, Standard Deviation. ^a^ Data are presented as number (percentage) of study participants unless otherwise indicated. Note: USD 1 is nearly equal to RMB 7. Note: *p*-values for the differences in baseline variables by quartiles of LCD score and its subtypes.

**Table 2 nutrients-14-01406-t002:** The association of low-carbohydrate-diet (LCD) score with all-cause mortality.

Characteristic	Quartiles of LCD Scores				*p* for Trend	*p* for Non-Linear
	1	2	3	4		
Overall LCD score ^a^						
Median score (IQR)	6 (4, 8)	13 (11, 14)	18 (17, 19)	24 (22, 26)		
Person-years of follow-up	74,195	77,724	71,289	71,640		
Mortality rate (per 1000 person-years)	182.1	156.3	140.1	147.8		
Crude HR (95% CI)	1.00	0.87 (0.80–0.94) *	0.79 (0.72–0.85) **	0.83 (0.77–0.90) **	<0.001	0.07
Adjusted HR (95% CI) ^d^	1.00	0.92 (0.85–0.99) *	0.89 (0.82–0.96) *	0.96 (0.88–1.04)	0.17	0.008
Adjusted HR (95% CI) ^e^	1.00	0.95 (0.87–1.02)	0.92 (0.85–1.02)	0.99 (0.91–1.08)	0.73	0.04
Adjusted HR (95% CI) ^f^	1.00	0.95 (0.88–1.03)	0.92 (0.84–1.00)	0.97 (0.89–1.06)	0.38	0.08
Vegetable-based LCD score ^b^						
Median score (IQR)	11 (9, 12)	14 (13, 15)	17 (16, 18)	20 (19, 21)		
Person-years of follow-up	77,358	85,584	77,518	54,388		
Mortality rate (per 1000 person-years)	151.0	148.2	161.8	171.7		
Crude HR (95% CI)	1.00	0.98 (0.91–1.07)	1.08 (1.00–1.17) *	1.16 (1.06–1.26) **	<0.001	0.15
Adjusted HR (95% CI) ^d^	1.00	0.99 (0.91–1.07)	1.12 (1.04–1.22) **	1.18 (1.09–1.29) **	<0.001	0.23
Adjusted HR (95% CI) ^e^	1.00	0.99 (0.91–1.07)	1.11 (1.02–1.21) *	1.16 (1.05–1.27) *	<0.001	0.18
Adjusted HR (95% CI) ^f^	1.00	0.98 (0.90–1.06)	1.09 (1.00–1.18)	1.09 (0.99–1.20)	0.01	0.20
Meat-based LCD score ^c^						
Median score (IQR)	6 (3, 8)	13 (11, 14)	19 (17, 20)	24 (23, 27)		
Person-years of follow-up	74,816	75,838	79,936	64,259		
Mortality rate (per 1000 person-years)	186.9	155.7	143.2	140.1		
Crude HR (95% CI)	1.00	0.84 (0.77–0.90) ***	0.78 (0.72–0.84) ***	0.77 (0.71–0.84) ***	<0.001	0.15
Adjusted HR (95% CI) ^d^	1.00	0.89 (0.82–0.96) **	0.87 (0.80–0.94) **	0.88 (0.81–0.96) **	0.001	0.12
Adjusted HR (95% CI) ^e^	1.00	0.89 (0.83–0.97) **	0.90 (0.83–0.97) **	0.89 (0.81–0.97) **	0.007	0.06
Adjusted HR (95% CI) ^f^	1.00	0.89 (0.82–0.97) **	0.91 (0.84–0.98) *	0.91 (0.83–0.99) *	0.01	0.07

Abbreviations: IQR = Interquartile Range; ^a^ = Low carbohydrate, high total fat, and high protein intake; ^b^ = Low high-quality carbohydrate, high unsaturated fat, and high plant protein intake; ^c^ = Low low-quality carbohydrate, high saturated fat, and high animal protein intake; ^d^ = Adjusted for sex and age; ^e^ = Additionally adjusted for education, family income, smoking, drinking, physical activity, BMI and history of cancer and CVD; ^f^ = Additionally adjusted for systolic blood pressure, fasting plasma-glucose, total cholesterol and self-rated health at baseline. *: 0.05; **: 0.01; ***: 0.001.

## Data Availability

Due to ethical restrictions protecting patient privacy, data available on request from the Guangzhou Biobank Cohort Study Data Access Committee. Please contact us at gbcsdata@hku.hk for fielding data accession requests.

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
