# Peer review of "Low-Carbohydrate Diets and Mortality in Older Asian People: A 15-Year Follow-Up from a Prospective Cohort Study"

_nutrients, 2022, doi:10.3390/nu14071406_

Round 1

Reviewer 1 Report

Thank you for the opportunity to review the manuscript “Low-carbohydrate diets and mortality in older people: a 15-year follow-up from a prospective cohort study”. The manuscript investigates the long term effects on LCD on mortality, taking into account carbohydrate quality and sources, in an elderly Asian cohort.

Overall, with the limitations outlined below, I think this paper is well written and constitute a quite sound scientific work. The results of this study are in the opposite direction to those expected in western population, due to a diet, specific to Asian older people, which is characterized by different proportions of macronutrients. This study contributes to the better understanding of the effects of LCD and more generally of carbohydrates on human heath.

Please see my further comments below.

-Title

Line 1: in the title precise “Asian older people”. Indeed effects observed in this study are characteristic of Asian diet and cannot be transposed to the Western diet.

-Abstact

Line 31: please add “Asian” in “older people”

-Introduction

Line 59: please justify why you were interested particularly in this cohort of older people. Better chance to observed death ? Well documented cohort ? Etc.

-Methods

Line 135: why this reference here Castro-espin 2021 ?

Line 144 to 149: It would be helpful to make this part clearer to readers who are not familiar with this method.

Line 153: could you explain why excluding participants who died within the first three years of follow-up account for potential bias due to reverse causality ?

There are no more line numbers from results section which make specific comments harder to write

-Results

A general comment for this section: there are too many tables and figures in the main text and in the supplementary part. Due to this, it is difficult to the reader to have access to all the information contained. Manuscript could gain with a lighter result part. Maybe, you could move Table.3 in the supplementary part. Bold characters for significant results in tables could also be appreciable.

-Discussion

There are several lacks and inaccuracies in the discussion section.

This section could be developed especially concerning the variability of LCD effects according to the specific needs of populations depending of local diet or age etc. I suggest this could be done either page 18 Line 3 or at the end of page 18.

A discussion on food pattern could also be an added value.

Part “In addition, the non-significant association between meat-based LCD and CVD mortality in diabetic...polyunsaturated fatty acids, 6.3% versus 8.2%)” is unclear.

Be careful with the comparison with previous studies : age in the studies you cited are different than in your cohort.

Concerning reference 20 (end of page 18), quality of carbohydrates was not controlled. You should mentioned it.

-Conclusion

I think it is important that in your conclusion you insist on the cause of the inconsistencies between the studies : inconsistencies due to the age difference between studies (in link with specific diet needs of each age) but also inconsistencies due to differences between local diet (i.e. US vs Asia).

I suggest you this sentence: “… with all-cause and CVD mortality in Asian older people. Inconsistencies in the literature on the health effect of LCD may reflect the importance of the local diet specificity and age-related requirements in composition of the diet.”

Reviewer 2 Report

Very interesting study. I congratulate the authors for conduction such an excellent study. There are minor suggestions in the attached document. The authors may want to consider them.
